# Assessment of Stress Salivary Markers, Perceived Stress, and Shift Work in a Cohort of Fishermen: A Preliminary Work

**DOI:** 10.3390/ijerph19020699

**Published:** 2022-01-08

**Authors:** Roberto Zefferino, Francesca Fortunato, Addolorata Arsa, Sante Di Gioia, Gianfranco Tomei, Massimo Conese

**Affiliations:** 1Department of Medical and Surgical Sciences, Faculty of Medicine and Surgery, University of Foggia, Via Napoli 121, 71122 Foggia, Italy; francesca.fortunato@unifg.it (F.F.); a.arsa@libero.it (A.A.); sante.digioia@unifg.it (S.D.G.); massimo.conese@unifg.it (M.C.); 2Department of Human Neurosciences, Faculty of Medicine and Surgery, University of Rome “La Sapienza”, Piazzale Aldo Moro 5, 00185 Rome, Italy; gianfranco.tomei@uniroma1.it

**Keywords:** stress, cortisol, immuno-mediated disease, cancer, melatonin, interleukin-1β

## Abstract

Due to work-related stress, today, work itself represents a daily challenge that must be faced in many occupations. While, in the past, the scientific community has focused on the helping professions, since, an increasing number of professions have since been investigated. Therefore, different approaches exist in order to assess this disorder, representing a scientific field wherein biological and psychological dimensions both need to be evaluated. In this paper, we consider three biological salivary markers: interleukin 1 beta (IL-1β), cortisol, and melatonin. The choice derives from recent contributions to the literature in which the interplay between these markers has been verified. Briefly, such interplay could explain how the central nervous, endocrine, and immune systems communicate with each other, supporting a holistic concept of person. In 30 marine fishermen from the Apulia region of Italy, perceived stress was measured using the Professional Stress Scale (PSS) and sleep disturbances were assessed through the Pittsburgh Sleep Quality Index (PSQI). Salivary markers were collected at 8:00 a.m. and 2:00 p.m. Those subjects reporting sleep disturbance and having altered scores in two PSS subclasses, home–work conflict and self-esteem, presented inverted salivary melatonin and cortisol nictemeral rhythms (with regard to melatonin levels at 8:00 a.m., those workers reporting values higher than the median showed 64.1% versus 48.6% home–work conflict with respect to cortisol levels, subjects having an inverted circadian rhythm showed 69.9% versus 52.5% home–work conflict, and these values resulted 47.7% versus 25.3% when the self-esteem was considered). As regards melatonin, PSQI score is statistically different in the two groups of subjects as identified by median melatonin at 8:00 a.m.; specifically, the subjects who had mean values higher than the median shared higher PSQI scores (10.8 versus 9.8). The same subjects reported more frequent home–work conflict and more sleep disorders. We found a negative correlation between IL-1β at 8:00 a.m. and Cortdiff (the difference between cortisol at 8:00 a.m.–cortisol at 2:00 p.m.), and that high IL-1β at 8:00 a.m. was associated with low Cortdiff. Based on our results we would like to propose this approach in health surveillance, in order to prevent mental and/or physical disorders, however our study is surely preliminary. The interesting perspectives and hypotheses cited in this paper, in which the roles of IL-1β and norepinephrine appear central and important, could remain hypothetical if not supported by more robust observation in order to produce, truly, new knowledge. In the future we will deepen this study with a larger sample, and if these results will be confirmed, this approach could allow preventing, not only mental and physical disorders, but also immuno-mediated diseases, and, perhaps, cancer.

## 1. Introduction

Job stress is a serious occupational risk in various tasks. It is known to determine mental disorders and/or physical diseases. Today, workers increasingly have to cope with great challenges. Usually stress affects the human service professions (e.g., teachers, social workers, health professionals), however, other jobs can also induce stress, which, in turn, has an impact on health. The first observations about stress were made by Selye, rightly considered the “father” of stress. He described stress as general adaption syndrome (GAS), specifying the role of the hypothalamic–pituitary–adrenocortical axis (HPA) and sympathetic–adrenal–medullary (SAM) systems. According to his theoretical construct, the two first phases—alarm reaction and resistance—could not appear as particularly harmful to health, while the third phase, which he named exhaustion, must be considered differently [1]. In fact, repetitive stresses appear insidious because they could exceed the ability to be coped with. Recent attention has focused on the causes of stress, considering subjective and objective aspects, which are both evaluated as important. Stress can derive from objective aspects, such as increased job demand, or can derive from neutral situations that the worker perceives as stressful. It may be useful to report the stress definition from S. Cohen: “The experience of negative events or the perceptions of distress and negative affect that are associated with the inability to cope with them” [2] This sentence explains that negative events are important, but also depend on perceptions of them, which can differentiate an event’s stressfulness along with individual responses to stressors.

Karasek and colleagues have considered the two aspects of job demand and job control [3]. In their opinion stress is dangerous when job control is low and job demand is high. The problems of this model are that it appears a bit too mechanical and that it is difficult to measure job control. More recently, the subjective factors of stress, identifying several mechanisms able to explain the “reversed effects” of mental health on work characteristics, have been considered [4]. However, in this study the weight of these factors are overrated. A different approach is used by Dikkers and colleagues, who demonstrate that workload appears as not simply causative of work–home interference, but as a potential consequence thereof, as well [5]. Another approach postulates that individual factors play a significant role in shaping the stress process, particularly coping capacity [6,7].

It is noteworthy that some approaches combine extrinsic and intrinsic factors within an integrative model. For instance, the “job demands-resources model” posits that personal resources are able to influence working people’s wellbeing along with the characteristics of the work environment. “Effort-reward imbalance” is another theoretical concept that combines extrinsic and intrinsic factors in studying work-related health [8]. It focuses on the work contract; particularly, it is based on the principle of social reciprocity. In fact, it considers rewards received in return for efforts expended, including money, esteem, and career opportunities. The lack of this reciprocity generates strong negative emotions and stress responses with adverse long-term effects on health.

Siegrist and Li [9] have proposed a new factor affecting occupational stress, which they have named worker–occupation fit (WOF), and have constructed a theoretical model of the effects of WOF on occupational stress and related disorders. WOF is defined in accordance with the theoretical concept of occupational stress as the match between a worker’s characteristics, needs, and abilities and the culture, supplies, and demands of the occupational environment. Furthermore, reporting the association between WOF and occupational stress, it has been shown that a lower level of WOF is associated with a higher level of occupational stress, indicating that occupational stress has a strong negative correlation with the level of WOF [10].

A causal factor linked to stress is fatigue, which could be considered, in turn, an independent risk factor able to increment injuries, as cohort studies would indicate [11].

Astrand and colleagues are the first to have studied work stress in fishermen [12]. A relevant definition of fatigue comes from the International Maritime Organization’s (IMO) guidelines on fatigue, defined as follows [13] “A reduction in physical and/or mental capability as the result of physical, mental, or emotional exertion which may impair nearly all physical abilities including: strength; speed; reaction time; coordination; decision making; or balance.” It is subsequently noted that fatigue, even in fishing, is the largest single contributing factor to accidents, thus, this area of research appeared highly under-prioritised to the authors [14]. These authors have examined five papers, providing an overview of research conducted on fatigue in fishermen and they concluded that “greater understanding is also needed to assess how much of the variance in fatigue is attributable to e.g., length of trip, hours of work without rest, and type of job and specific tasks”. Allen and colleagues examined 81 British fishermen and observed that 13 (16%) had been involved in a fatigue-related accident or incident, 36 (44%) said they had worked to the point of exhaustion or collapse, 33 (41%) had fallen asleep at the wheel, and 34 (43%) had been so tired they had slept on the deck or in the gangway [13].

Gander and colleagues studied the sleep and sleepiness of 20 fishermen and observed important differences during the peak of the hoki fishing season, comparing the last three days at home and the first three days at sea [15]. In an attempt to objectively assess stress, an increment of serum cortisol and prolactin without alterations of urinary catecholamines were found in “deep-sea” fishermen [16].

In a recent work [17], we proposed a model capable of explaining how melatonin, IL-1β, and cortisol interact during chronic stress (Figure 1). The choice of these markers depends on their different roles during chronic and acute stress. In this regard it is important, briefly, to remember that during acute stress cortisol increases to prepare individuals “to fight or to flight” [18]. Salivary cortisol variations have been indeed studied in relation with psychosocial work stress, indicating a normal, healthy response to work stress in most workers may be because many psychosocial work stressors were of mild or moderate intensity [19]. Regarding IL-1β, its increase was proved during stress [20]. Interestingly, there is evidence that cytokines, including IL-1β, are involved in physiological sleep regulation and have strict relationships with other sleep-regulatory substances (SRSs) [21,22,23]. Moreover, it was demonstrated that IL-1β possesses the ability to activate the HPA axis and suppress the hypothalamic–pituitary–gonadal (HPG) axis [24,25,26]. Hence, O’Connor and colleagues, underlining these neuroendocrine effects, proposed its role in homeostatic adaptation during an immune challenge [27]. These effects are produced by IL-1β acting on specific brain regions regulating the HPA and HPG axes.

Melatonin is considered not only a pineal gland-derived neurohormone implicated in the regulation of sleep but also a substance with pleiotropic effects, especially on the immune system [28]. Noteworthy, the alterations of these markers occurring during chronic stress are very different with respect to acute stress [29]. We hypothesized [17] that during chronic stress the interplay between melatonin, cortisol, and IL-1β is capable of triggering a loop that, through cortisol resistance, increments IL-1β, configuring an imbalance of cytokine concentrations toward a pro-inflammatory state. This state is very harmful for health; particularly, it associates with mental, immuno-mediated, and cardiac diseases, and perhaps cancer [30]. In a state of glucocorticoid resistance, the increase of IL-1β should transform this inflammatory signal into a nervous one, with heightened production of norepinephrine (NE). In turn, NE uses the endocrine system (melatonin and cortisol) to counterbalance IL-1β, but these controls do not operate under cortisol resistance [17].

Within this conceptual frame we considered a particular line of work, fishing. This occupation is considered one of the most dangerous. Features common to the fishing occupation include exposure to cold, wind, rough seas, substantial participation of physical effort, frequent injuries during work, unpredictable and abrupt threats [31], and, importantly, work-related stress.

Specifically, we aimed at studying stress in fishermen, evaluating the complex link between the biomarkers of stress (cortisol, IL-1β), focusing on their circadian behaviour, perceived stress, and sleep quality. Thus, the relevant questions of this article are: (i) did the perceived stress investigated using a PSS test and the sleep disturbances evaluated through PSQI correlate with the biological markers used in this study? (ii) Did these biological markers exhibit behaviour capable of indicating their reciprocal correlation in this study? Furthermore, (iii) could sleep disturbances associated with stress, along with night shift work, alter the circadian rhythms of the above-cited biological markers?

Finally, as speculative background and perhaps a conclusive hypothesis, this study would like to highlight this complex interplay during chronic stress, which, in our opinion, is important in order to try to explain how life and/or work events could affect health by inducing stress-correlated diseases.

## 2. Materials and Methods

### 2.1. Subjects

This paper proposes a preliminary study of 30 fishermen belonging to several fishing companies from the Apulia Region of Italy. Informed consent was obtained from all volunteer subjects. They were provided with adequate information concerning the study and provided with adequate opportunity to consider all options. We responded to all subjects’ questions, ensuring that each subject had comprehended this information, then obtained their voluntary agreement. All thirty fishermen underwent a medical examination, including the collection of medical histories that permitted us to know the demographic and clinical information (i.e., smoking status, health etc.) of the whole sample.

The descriptive characteristics of the sample considered in this study are illustrated in Table 1.

### 2.2. Evaluation of Perceived Stress

The three-shift system is the most common plan for five, 24-h days per week. The “first shift” often runs from 06:00 to 14:00, the “second shift” or “swing shift” from 14:00 to 22:00 and a “third shift” or “night shift” runs from 22:00 to 06:00, but shifts may also have different lengths to accommodate workload, e.g., 7, 8, and 9 or 6, 8 and, 10 h. We measured perceived stress using the PSS test [32], a psychometric test that explores stress determinants considering six subclasses: workload, lack of organization, lack of resources, conflicts, self-esteem, and home–work conflict. Higher scores correlate with a specific stress subclass. Sleep disorders were studied by using the Italian form of the Pittsburgh Sleep Quality Index (PSQI) [33]. The PSQI is a useful instrument, able to measure quantitative and qualitative sleep features [34]. It consists of a 19-item self-reported questionnaire capable of evaluating sleep quality over the preceding month. It investigates seven clinically-derived component scores, considering sleep quality, sleep latency, sleep duration, habitual sleep efficiency, sleep disturbances, use of sleep medication, and daytime dysfunction. Higher scores indicate worse sleep quality [35]. A total score value of five or higher indicates an alteration of sleep quality. The PSQI’s sensitivity is 90.0%, its specificity is 67.0% [36].

### 2.3. Saliva Samples Collection

The participants were informed about the correct collection method of salivary sampling and were warned that they had to chew the cotton swab for at least 5 min (Salivette, Sarstedt, Germany).

They were asked to collect two saliva samples at 8:00 a.m and 2:00 p.m. and not to eat or drink (except for water) or brush their teeth within 1 h prior to sample collection, in order to minimize possible food debris and the stimulation of salivation. Salivary samples, after collection, were brought to the laboratory and stored frozen at −20 °C, until required for assays.

### 2.4. ELISA Assays

In order to verify and quantify the biological effects of chronic stress, we evaluated the salivary concentrations of cortisol, IL-1β, and melatonin. As such, every worker collected two salivary samples, one at 8:00 a.m. and one at 2:00 p.m. Samples were stored below −20 °C until quantification. ELISA kits were used for the determination of cortisol, IL-1β, and melatonin (Salimetrics, LLC, Carlsbad, CA, USA). The assay sensitivities were <0.007 μg/dL, 0.37 pg/mL, and 1.37 pg/mL for cortisol, IL-1β, and melatonin, respectively.

In the Results Section, we will use the following terms:

Cortisol difference = cortisol at 8:00 a.m.–cortisol at 2:00 p.m. (Cortdiff); IL-1β difference (IL1diff) and melatonin difference (Melatdiff) = difference between 2:00 p.m. value and 8:00 a.m. value.

### 2.5. Statistical Analysis

Continuous variables with normal distribution were compared by Student’s *t*-test. Alternatively, the non-parametric Kruskal–Wallis test was used. To assess the distribution of the variables, Bartlett’s test was used. To evaluate the association between qualitative variables, univariate analysis was performed using double-entry contingency tables and computing chi-square and odds ratio with 95% confidence intervals, with a significance level of ≤0.05.

Spearmen’s rho was measured to verify the strength and direction of the association of two variables (correlation), then *p* was calculated to ascertain the statistical significance of this association. The observed differences were considered statistically significant if *p* was <0.05. The analysis was conducted with Stata/Se 15.0 (StataCorp LLC, College Station, TX, USA).

## 3. Results

### 3.1. Salivary Marker Levels

Descriptive data about salivary markers, including measures of central tendency and dispersion around the mean, are shown in Table 2.

### 3.2. IL-1β

IL-1β is low in normal subjects in the morning, and, like melatonin, increases during the day; vice versa, cortisol nictemeral rhythm is characterized by an increase of cortisol concentration in the early morning followed by its value dropping over the course of the day, determining a high Cortdiff. In the fishermen cohort, we found a negative correlation between IL-1β at 8:00 a.m. and Cortdiff (Figure 2a). In particular, in the left part of Figure 2a we can observe subjects that have high differences of cortisol and low IL-1β levels. These subjects are healthy because, at 8:00 a.m., IL-1β is normally lower; in fact, if subjects have no sleep disturbance, IL-1β normally decreases during sleep. Thus, its value should appear low at 8:00 a.m. Instead, in the right part of Figure 2a, we may note those subjects showing higher levels of IL-1β at 8 a.m., whereas, at the same time, they share minimal circadian cortisol variability, indicating an alteration of both markers that could affect their health. In fact, they might have cortisol resistance, since, during this state, an amplifying loop may occur and induce an increase in IL-1β [17].

Figure 2b shows that subjects which smoked more cigarettes had more negative IL1diff, indicating that cigarette smoking could increase the production of IL-1β and invert its circadian variability.

No significant associations were identified between IL1diff, the PSS test, and PSQI scores.

### 3.3. Melatonin

From Figure 3a we can observe that subjects who had higher melatonin at 8:00 a.m. perceived less lack of resources (low scores on the PSS test). In Figure 3b we can observe that subjects who had normal melatonin variability showed higher scores in lack of organization, another subclass of the PSS test. Figure 3c shows that subjects with lower Melatdiff had also lower PSQI scores.

Table 3 reports that the subjects showing higher melatonin at 8:00 a.m. report often/always putting “effort into falling asleep”, such as premature awakening, and these levels were statistically significant as compared with never/sometime responders. Considering Melatdiff, the difference between those who report often/always premature and night awakening and those which experienced never/sometime this problem was significant, indicating that they have inverted circadian rhythms (*p* < 0.05).

The results presented in Table 4 point out that the percentage of subjects showing a melatonin value at 8:00 a.m. higher than the median report more frequent home–work conflict. This result is in agreement with the results shown in Table 3, where inverted melatonin rhythm (negative Melatdiff) is associated with premature and night awakening. In Table 4 we can observe that the mean of the Pittsburg Test scores is statistically different in the two groups of subjects identified by median of melatonin at 8:00 a.m.; particularly, the subjects that had mean values higher than the median shared higher scores. Overall, these results indicate that sleep disturbance was capable of increasing melatonin at 8:00 a.m. when it should normally be very low, thus denoting that a serious sleep disturbance could be capable of determining an inversion of this rhythm or vice versa.

### 3.4. Cortisol

When we divided the fishermen into two groups by considering the median value of Cortdiff, the subjects who had lower levels than the Cortdiff median showed wider sleep latency, higher scores in the subclasses of PSS, such as home–work conflict, and in self-esteem, differences that are statistically significant (Table 5). Moreover, subjects who presented low concentrations at 8:00 a.m. (altered value) have more psychiatric disorders, particularly depressive disorders, but these difference do not appear statistically significant (data not shown).

In Figure 4a we can verify the inverse correlation between home–work conflict and Cortdiff: those subjects who showed altered PSS test scores for home–work conflict had low Cortdiff. These results show that subjects having minimal cortisol circadian variability perceive higher stress in the subclass home–work conflict, confirming that conflictual situations might provoke a decrease of cortisol variability and the development of glucocorticoid resistance.

Figure 4b shows subjects who have higher scores on the PSQI have higher cortisol variability than the subjects that have lower scores on the PSQI.

### 3.5. Shift Work

In our sample the workers had the following shifts: 85% took only night shifts for five days (Sunday–Monday until Thursday–Friday), 15% took both first shifts and night shifts. We considered the difference between subjects that worked only at night, i.e., night-shift workers (NSW) and subjects that worked both first shifts and night shifts, i.e., both-shift workers (BSW). This evaluation yielded evidence that BSW have higher variability of cortisol and length of sleep than NSW. On the other hand, NSW have higher stress in the subclass of home–work conflict, higher systolic blood pressure, more night and premature awakening, and more difficulty falling asleep (Table 6).

## 4. Discussion

Research on occupational stress might represent an essential chance for developing preventive and promotional strategies oriented toward mental and physical well-being. From this perspective, this study aims to examine the relationship between perceived occupational stress and the salivary markers of chronic stress. Moreover, an important aspect, the presence of the sleep disturbances, was investigated as well, without neglecting the shiftwork influence. Studying the link between the occurrence of psychological perceptions using methods that identify perceived stress and biological markers could allow grasping the biological features of work stress, and would prove useful in the prevention or early detection of disorders that are able to develop into real and true illness.

The use of saliva is useful because its collection is simple and not at all invasive; moreover, the collection method used by us—chewing—increases the concentrations of the majority of metabolites, and the stimulation of flow rate induced by mastication of cotton swab is particularly useful in research utilizing salivary samples, as it is known that citric acid stimulation reduces the concentrations of metabolites [37,38].

This preliminary study shows that Cortdiff and IL-1β at 8:00 a.m. are negatively correlated. It is well known that IL-1β at 8:00 a.m. is low in subjects reporting good quality of sleep [21,23]. In our sample, low concentrations of IL-1β at 8:00 a.m. are associated with normal cortisol variability, in good agreement with the literature [21,22,23], showing again that good quality of sleep acts as anti-inflammatorily. Noteworthy, our study shows the association of a slight variability of cortisol with high IL-1β levels at 8:00 a.m., giving evidence that sleep deprivation might be considered a pro-inflammatory agent, in agreement with previous literature [21,22]. Recently, several authors have demonstrated that the innate immune system response can be activated in particular social environments; in fact, social conflict, evaluation, rejection, isolation, or exclusion, perhaps due to the implications these conditions historically had, could be considered as particularly harmful for mental and physical health. Several authors showed that there exists an ancestral host defence mechanism activated by nonphysical, social, symbolic, anticipated, or imagined threats. In turn, this mechanism can increase an individual’s risk for both viral infection and inflammation-related disease [17,39,40]. Moreover, it has been proposed [41] that chronic stress can induce a characteristic clinical condition named “chronic stress-induced GC resistance”, which is able to dampen anti-inflammatory processes and induce the prolonged production of pro-inflammatory mediators. In this regard, an important effect seems to be played by cortisol due to its known effect on the immune system. This was investigated by Chrosus and Elenkov, who verified that cortisol was able to polarize naive CD4+ T cells toward the Th2 subset [42]. Particularly, this polarization makes subjects more susceptible to infective disorders and autoimmune diseases, as well as less reactive toward cancer. Particularly useful is to remark upon the role of HPA axis hyperactivity and the resulting GC (glucocorticoid) resistance; it might represent the link between chronic stress effects and major depressive disorder (MDD), diabetes, or metabolic syndrome [43].

However, it is useful to add that, because of glucocorticoid resistance, HPA axis mechanisms that support adaptive “fight or flight” responses to social–environmental threat can be altered, promoting excessive inflammation, particularly when they appear either frequent or chronic. These mechanisms could explain the link between mental and physical well-being [44,45,46,47].

In our study, certainly preliminary, we observed that subjects with minimal cortisol circadian variability perceived higher stress in two subclasses of PSS, home–work conflict and self-esteem. This confirms that conflictual situations might provoke decreasing cortisol variability that could be linked with a cortisol-resistant state [48].

The results regarding shift work demonstrate that BSW have higher variability of cortisol and longer sleep than NSW, whereas these latter perceive, as more important, stress in the subclass of home–work conflict and have higher systolic blood pressure, more numerous night and premature awakening, and more pronounced difficulty falling asleep, confirming the literature data [49,50,51,52].

The negative correlation between IL-1β at 8:00 a.m. and Cortdiff highlighted that those subjects who have higher levels of IL-1β share minimal circadian cortisol variability, which could be harmful to their health. Indeed, they might have cortisol resistance, since it is known that during cortisol resistance an amplifying loop occurs that is capable of increasing IL-1β [17]. We also show that subjects who smoke more cigarettes have more negative IL1diff, indicating that cigarette smoking could increase the production of IL-1β and could be able to invert its circadian variability, a result in agreement with previous data [53]. Our results are in good agreement with a study that reported plasma cortisol decrease from morning to midday in relation to IL-6, showing a very clear negative relationship between a flat cortisol curve and high IL-6 and also demonstrating that a high IL-6 is related to low job control in men [54].

With regard to melatonin and PSQI score the subjects that had mean values higher than the median (at 8:00 a.m.) shared higher PSQI scores. The same subjects reported more frequent home–work conflict and more sleep disorders.

From the results shown in Figure 2a,b, we might hypothesize that subjects who have sleep disorders and increased melatonin at 8:00 a.m. would be less prone to analysing their work conditions. Conversely, the subjects who had normal melatonin variability could have also good analysis skills, explaining why they show higher scores in lack of organization, another subclass of the PSS test. On the other hand, subjects with lower Melatdiff also shared lower PSQI scores, suggesting that their lower circadian differences in melatonin might be considered, in this context, as an early alteration and not yet associated with PSQI high scores, which might occur later.

Subjects with high melatonin at 8:00 a.m. reported premature awakening, had home–work conflict, and more sleep disorders; this outcome corroborates previous reports in the literature [55,56], since, normally, at 8:00 a.m. melatonin concentrations should be very low. If we consider Melatdiff, we observe that subjects with inverted circadian rhythm show premature and night awakening and report more nightmares than normal subjects. In fact, it is known that melatonin increases following long periods of exercise, especially if the body continuously works during the normal period of night time sleep, inducing altered circadian rhythms [57]. Several authors observed that either cognitive function or core temperature and melatonin production are altered during sleep deprivation in tandem with prolonged exercise [58,59,60]. In line with these reports, other authors showed that sleep deprivation increases melatonin release [55,56] and that a direct proportionality between melatonin release and the duration of sleep deprivation exists; in fact, the longer sleep the deprivation, the greater the melatonin release and drop in body temperature [61].

This response to sleep deprivation is likely a physiological defence mechanism that could induce humans to sleep in order to regain normal physiological function [58]. However, in turn, high concentrations of melatonin were shown to increment IL-1β levels in mice splenocytes [62]. Wichman et al. demonstrated that low levels of IL-1 and IL-6 production, observed in mice following experimental trauma-haemorrhage, returned to basal control levels after melatonin treatment [63]. Nevertheless, melatonin showed alternate effects; in fact, during rat heatstroke-induced lung inflammation and airway hyper-reactivity, it reduced neutrophil infiltration and levels of inflammatory mediators [64,65]. Moreover, melatonin was capable of contrasting the increased production of cytokines, in numerous in-vivo models of inflammation [66,67,68,69,70,71,72,73,74,75,76].

There exists a diurnal rhythmicity to human cytokine production; in fact, the IFN-γ/IL-10 peak occurs during the early morning, which is positively correlated with plasma melatonin [77], suggesting an important role played by melatonin in determining Th1 polarization. Oppositely, several in-vivo studies have reported that melatonin can promote a Th2 response in different models. Particularly, melatonin, at high doses, was capable of incrementing the production of the cytokine IL-4 (a hallmark of Th2 response) in bone marrow lymphocytes [78]. During early nocturnal sleep, a shift in the Th1/Th2 cytokine balance occurs towards increased Th1 activity; instead, the Th2 response prevails during late sleep. Dimitrov et al. observed an evident decrease in TNF-α-producing CD8+ cells during sleep [79], indicating a possible correlation between melatonin and Th2 response. In rats, after pynealectomy a polarization of Th1/Th2 cells towards Th1 response was demonstrated, giving proof that melatonin directs the immune response towards Th2 dominance [80].

Our study failed to show a direct correlation between melatonin and IL-1β; if we plot together their concentrations in order to verify a correlation, perhaps because the workers were in a phase of stress incapable of verifying this alteration, we must consider stress as a condition, in that the markers change over time, following its different phases. Another explanation could be an indirect pathway that considers NE, as is considered below.

The involved mechanisms in different stress phases are unclear, but the above observations might indicate a possible molecular link between sleep, inflammatory response, and immunity. Additionally, stress and the immune system are interconnected and molecular links were identified between cortisol, norepinephrine, melatonin, and IL-1β. Here, we could affirm to have shown as much, using salivary markers, the PSS test and the PSQI, as they might be linked to each other. Emphasizing once again that our data are preliminary, we might add that melatonin is associated with sleep disturbances (nocturnal and premature awakening and nightmares), as evaluated with the PSQI and by home–work conflict, itself evaluated through the PSS test. Furthermore, we observed low cortisol variability associated with work–home conflict and self-esteem as well as a wider sleep latency and an increment of IL-1β at 8:00 a.m. It is useful to remark that our study is a preliminary study, and we do not have a control group, although each subject is the control of himself. These alterations are also, in part, verified after having divided the samples according to shift type.

Another limit of our study is that few subjects were considered. Here, we are facing complex systems, thus only an appropriate sample could permit verification of the roles of different variables. As we have recently reported [17], we tried to verify theoretically a possible interplay between different markers during chronic stress. Nevertheless, we retain that our study might indicate an objective method of ascertaining work-related stress that considers perceived stress and biological markers of stress simultaneously; moreover, this method might be useful in identifying subjects that have a greater risk of developing autoimmune diseases and/or alterations able to affect mental and physical wellness, and of developing sleep disorders that appear capable of impacting the immune system and psychological wellbeing. This possibility, presently, is only a speculative research line that needs to be ascertained using a larger sample, therefore it will also be useful to have a suitable control group, even though it is noteworthy that each individual provided two samples and that this method reduced the inter-individual variability when differences were analysed. Finally, from a mechanistic point of view, given the important counterbalancing role that the HPG axis plays in IL-1β-mediated effects, in the future, a detailed look at the HPG axis is deserving of study concerning work-related stress.

According to the above results we can answer the introductive questions, verifying some of the founding hypotheses from our previous work. In particular, we find that (i) fishermen with low Cortdiff also show higher levels of IL-1β at 8:00 a.m. and (ii) an increase of melatonin at 8:00 a.m. and an inversion of its CD rhythm are both associated with higher PSQI score (Figure 5). It remains how to link the increase of both IL-1β and melatonin at the same time, considering that the increase of IL-1β associates with the decrease of melatonin, as shown at the bottom of Figure 5. However, there exists another pathway capable of increasing melatonin following the increase of IL-1β; in particular, as shown in the central part of Figure 5, higher levels of IL-1β result in the rise of NE, which, in turn, increases melatonin. Therefore, it may be that this pathway is more powerful than the mere direct inhibitory effect of IL-1β on melatonin, especially if we consider that, during chronic stress, cortisol increases NE, too, with very high efficiency [81].

## 5. Conclusions

Taking the appropriate distance from the possibility of explaining the complex mechanisms of stress, and only considering the studied sample, we can nonetheless draw some conclusions. Firstly, fishing is a dangerous occupation due, especially, to night-shift work. Secondly, night-shift work, in our sample, is associated with sleep disturbances. Thirdly, sleep disturbances are linked with an inversion of IL-1β, cortisol, and melatonin circadian rhythms. Finally, considering the assumed role of NE—not investigated in this study but provided by literature data—an interesting speculative hypothesis could explain the link between IL-1β and melatonin, opening new insights in the study of stress phenomenology.

We must remark that the study of the interplay of these complex biological markers during chronic stress needs to be deepened in order to explain how negative events in life and/or work—or those only perceived as such—could affect health by inducing stress-correlated disease. Finally, the study of this phenomenon deserves further deepening to further evidence our hypothesis, a challenge for the coming years.

## Figures and Tables

**Figure 1 ijerph-19-00699-f001:**
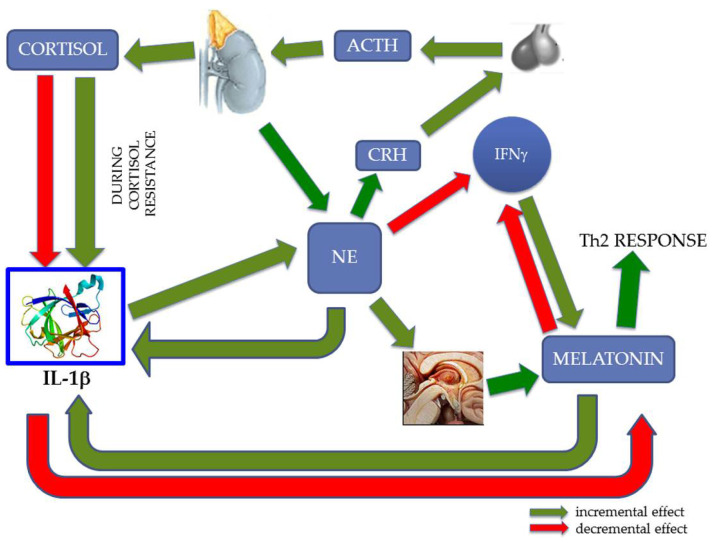
Interplay between central nervous system, endocrine system, and immune system. Under physiological conditions, norepinephrine (NE) activates the HPA axis and determines the production of cortisol that, in turn, downregulates the production of IL-1β. During chronic stress and especially a glucocorticoid-resistant state, the increment of IL-1β transforms this inflammatory signal into a nervous signal: NE. In turn, NE uses the endocrine system (melatonin and cortisol) to rebalance IL-1β, but this control cannot function under cortisol resistance. NE eventually induces a skewing of the immune response, with a reduction in interferon gamma (IFNγ) and T helper 1 (Th1) activation, and an unbalanced T helper 2 (Th2) response. Other abbreviations: corticotropin releasing hormone (CRH); adrenocorticotropic hormone (ACTH). The protein structure of Interleukin 1β (IL-1β) was taken from the RCSB-protein data bank (www.rcsb.org, accessed date 15 February 2020 [17].

**Figure 2 ijerph-19-00699-f002:**
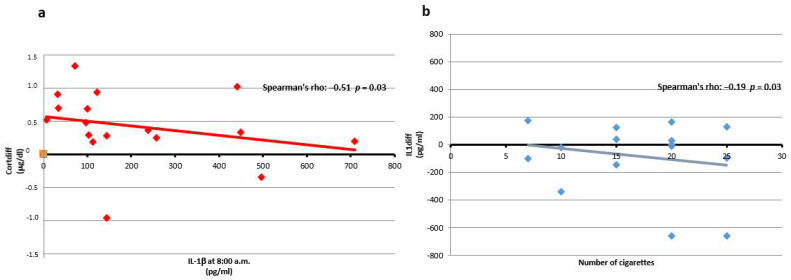
Correlations of IL-1β with Cortdiff and number of cigarettes. (**a**) The negative correlation between IL-1β levels at 8:00 a.m. (pg/mL) and Cortdiff (μg/dL) is shown with a Spearman’s rho = −0.51 and *p* = 0.03. (**b**) The negative correlation between IL1diff and number of cigarettes is shown with a Spearman’s rho = −0.19 and *p* = 0.03.

**Figure 3 ijerph-19-00699-f003:**
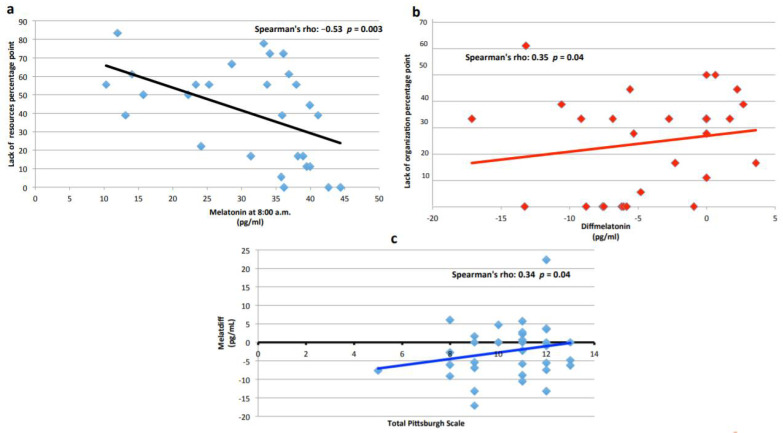
(**a**) Correlation of melatonin with subclasses of PSS and total PSQI score. The negative correlation of melatonin levels at 8:00 a.m. (pg/mL) and the PSS subclass “lack of resources” is shown with a Spearman’s rho = −0.53 and *p* = 0.003. (**b**) Positive correlation of Melatdiff (pg/mL) and the PSS subclass lack of organization is shown with a Spearman’s rho = 0.35 and *p* = 0.04. (**c**) The positive correlation of Melatdiff (pg/mL) and total PSQI score is shown with a Spearman’s rho = 0.34 and *p* = 0.04.

**Figure 4 ijerph-19-00699-f004:**
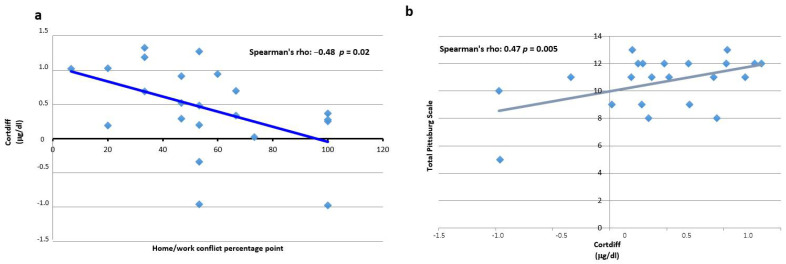
(**a**) Correlation of cortisol with the home–work conflict subclass and total PSQI score. The negative correlation between Cortdiff (μg/dL) and the PSS subclass home–work conflict is shown with a Spearman’s rho = −0.48 and *p* = 0.02. (**b**) The positive correlation between Cortdiff (μg/dL) and PSQI score is shown with a Spearman’s rho = 0.47 and *p* = 0.005.

**Figure 5 ijerph-19-00699-f005:**
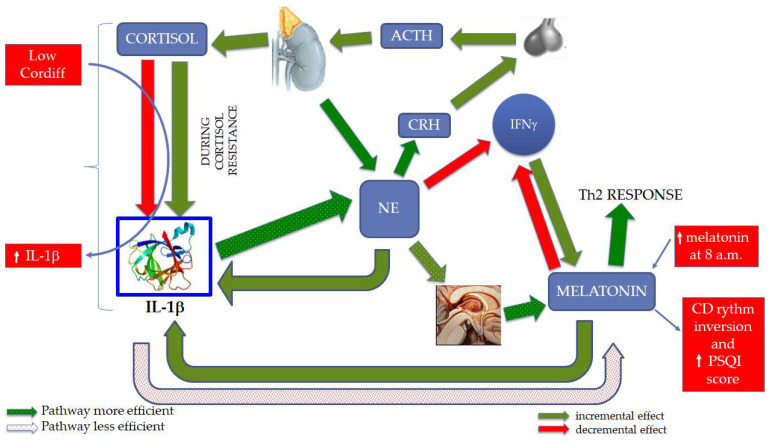
Verification of our hypothesis of the relationships between cortisol, IL-1β, and melatonin. Results obtained in a fishermen cohort subjected to chronic stress show that low Cortdiff is associated with higher IL-1β levels, while increased melatonin levels at 8:00 a.m. are associated with circadian (CD) inversion and higher PSQI scores, i.e., sleep disturbances. The dotted arrows hypothesize the relationship between increased IL-1β levels, NE, and melatonin occurring in fishermen, in particular the more efficient pathway stemming from IL-1β, increasing NE and, eventually, melatonin. This pathway would overcome the less efficient direct inhibitory pathway of IL-1β on melatonin. Other abbreviations: norepinephrine (NE); interferon gamma (IFNγ); T helper 1 (Th1); T helper 2 (Th2); corticotropin releasing hormone (CRH); adrenocorticotropic hormone (ACTH); interleukin 1β (IL-1β); circadian rhythm (CD); Pittsburgh Sleep Quality Index (PSQI); the difference between cortisol at 8:00 a.m.–cortisol at 2:00 p.m. (Cortdiff).

**Table 1 ijerph-19-00699-t001:** Descriptive characteristics of fishermen investigated in this study.

Characteristic	Mean	SD	CI (95%)	Range
age (years)	49.6	11.4	45.6–53.5	27–71
length of service (years)	30.6	12.9	26.2–35.1	6–50
systolic pressure (mm/Hg)	129	11.9	124.9–133.1	100–160
diastolic pressure (mm/Hg)	87.7	8.2	84.7–90.5	70–105
number of cigarettes	17.8	5.9	14.1–19.4	5–25

**Table 2 ijerph-19-00699-t002:** Descriptive statistical data about salivary markers levels.

Marker	Mean	Mean Difference	SD	Median	CV (%)
cortisol at 8:00 a.m.	1.23		0.48	1.39	39.0
cortisol at 2:00 p.m.	0.71	0.52	0.50	0.56	70.4
IL-1β at 8:00 a.m.	239.4		229.3	145.0	6.4
IL-1β at 2:00 p.m.	179.8	−59.6	183.6	124.0	6.8
melatonin at 8:00 a.m.	30.9		10.2	34.9	33.1
melatonin at 2:00 p.m.	28.0	−2.9	9.6	29.9	34.3

Mean and median values are in μg/dL for cortisol and in pg/mL for Interleukin 1 β (IL-1β) and melatonin; SD: standard deviation; CV: coefficient of variation.

**Table 3 ijerph-19-00699-t003:** Association of melatonin levels with sleep disorders.

	Never/Sometime		Often/Always		*p*
	Mean	SD	Mean	SD	
**Effort into falling asleep**					
higher melatonin (8:00 a.m.)	27.6	10.4	32.5	10.4	0.0351
**Premature awakening**					
higher melatonin (8:00 a.m.)	26.1	10.3	34.9	8.4	0.0093
Melatdiff	0.06	8.52	−5.42	6.54	0.0454
**Night awakening**					
Melatdiff	0.22	6.05	−5.02	9.19	0.0370

**Table 4 ijerph-19-00699-t004:** Association of melatonin levels with the subclass home–work conflict of the PSS and PSQI.

	Home/Work Conflict	
	Mean ± SD	*p*
melatonin (8:00 a.m.)		
<median	48.6 ± 16.2%	0.0414 *
>median	64.1 ± 29.6%	
	Total PSQI **	
	Mean	*p*
melatonin (8:00 a.m.)		
<median	10.8 ± 1.8	0.0386 *
>median	9.8 ± 1.7	

* Student’s *t*-test. ** PSQI = Pittsburgh Sleep Quality Index.

**Table 5 ijerph-19-00699-t005:** Association of Cortdiff levels with sleep latency, the PSS subclasses home–work conflict and self-esteem.

Cortdiff	Mean ± SD	*p*
	**sleep latency**	
<median	15.0 ± 8.0 min	
>median	8.7 ± 4.0 min	0.04 *
	**home/work conflict**	
<median	69.9 ± 27.0%	
>median	52.5 ± 24.0%	0.034 *
	**self-esteem**	
<median	47.7 ± 28.2%	
>median	25.3 ± 18.4%	0.01 *

* Student’s *t*-test.

**Table 6 ijerph-19-00699-t006:** Association of the shift system with perceived stress and biological and clinical parameters.

Shift System	BSW		NSW		
	Mean	SD	Mean	SD	*p*
Cortdiff (μg/dL)	1.17	0.21	0.30	0.55	0.01
length of sleep (hours)	8.6	0.89	5.0	0.87	<0.01
home–work conflict	25%	11.92	63%	23.62	<0.01
systolic blood pressure (mm Hg)	121	16.73	130	10.82	0.05
Effort into falling asleep					
often/always	20%		53%		0.02
never/sometime	80%		47%		
47% Night awakening					
often/always	40%		60%		0.03
never/sometime	60%		40%		
40% Premature awakening					
often/always	20%		63%		0.02
never/sometime	80%		37%		

Cortdiff (the difference between cortisol at 8:00 a.m.–cortisol at 2:00 p.m.). NSW: subjects that worked only at night (night-shift workers). BSW: subjects that worked both first shifts and night shifts (both-shift workers).

## Data Availability

The data that support the findings of this study are available from the corresponding author upon reasonable request.

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
