# Peer review of "Assessment of Stress Salivary Markers, Perceived Stress, and Shift Work in a Cohort of Fishermen: A Preliminary Work"

_ijerph, 2022, doi:10.3390/ijerph19020699_

Round 1
Reviewer 1 Report
Assessment of Stress Salivary Markers, Perceived Stress and Shift Work in a Cohort of Fishermen; A Preliminary Work
Zefferino R et al
Saliva is used as a source for diagnostics and prognostics related to perceived stress and sleep disorders. In addition they use a PSS test to correlate the perceived stress to the biological markers with five sub aims including sleep disturbances.
The introduction which review the research field, 3 pages , is comprehensive and leads to that norpinephrine (NE) is central in the interplay between central nervous system, endocrine system and immune system, however, not measured in this study.
Salivary samples taken by Salivette, at 8 am and 2 pm were examined for IL-1 B, melatonin and cortisol using Elisa assays. The assays sensitivities were presented.
The data were presented as differences between the two sample points.
The authors do not present either the obtained data nor the variation coefficients in the assays. Information of how these parameters differ during a day and a night is not included. Such data would have made it possible to evaluate the presented data. The correlation plots are not easy to interprete.
The discussion is also very comprehensive in order to explain these complex mechanisms. They conclude that fishery is dangerous working night shift due to sleep disturbances which may be linked to an inversion of IL-1 b, cortisol and melatonin; leading to a speculative hypothesis.
However saliva is an interesting biological source
Reviewer 2 Report
Dear Editor/Authors
Overall, this is an interesting study, focusing on various salivary biomarkers on a non-healthcare group working night shifts.
The authors studied the association between work activities, shift work effects and stress related responses in 30 fishermen as well as they assessed a number of salivary biomarkers. The authors concluded that sleep disturbances in fishermens are linked with an inversion of IL-1 , cortisol, melatonin circadian rhythms.
This is an interesting approach, but I have few concerns with this study, including adding some additional data:
- It is not described how exactly the collection of saliva was done although this represents a key moment. Was the salivary collection done by yourself or under the supervision of a healthcare professional? For how many minutes was the saliva collected? With regard to the salivary biomarkers tested, please relate the values found to the actual flow rates of the participant as many of these salivary biomarkers are flow rate dependent. Authors could add a discussion on the potential effect of flow rate dependency of the biomarkers for the results as well as that it should be clear for how long a period the saliva collection was done (e.g., 2 min, 5 min, 10 min).
- There is no mention of which medicaments the participants used. It is not clear whether the authors considered neurological or psychiatric disorders as potential contributing causes for the development of sleep disturbances.
The figures and tables are well structured and clear. There are no English writing matters, the text is easy to read and correctly written.
Thank you for the opportunity to evaluate this manuscript, which is, in my opinion, suitable for publication and is within the scope of the journal, after a minor revision.
Reviewer 3 Report
The general background omits (as most stress reviews still do, unfortunately) the important counterbalancing role that the HPG axis plays in all of this. Most researchers continue to measure only parameters that are related to the HPA axis and the sympatho-adrenomedullary system. But the authors cannot be blamed for this which is a weakness in the whole field.
Their literature references are in general ok but they do not seem to be aware of the fact that cortisol decrease has been studied extensively in the literature. Flat cortisol rhythm is an important parameter in
Kristenson et al (Bentham books) The Role of Saliva Cortisol Measurement in Health and Disease, 2012, 43-66 - which has a chapter on psychosocial work stressors and saliva cortisol
There has also been research on interleukins and cortisol regulation for a long time. There is for instance one publication on cortisol decrease (plasma, not saliva) from morning to midday in relation to IL-6 which is closely related to, behaves like and seems to have a role similar to that of IL-1beta in an epidemiological study from 2000. It shows a very clear negative relationship between a flat cortisol curve and high IL-6 (Stress Medicine 2000), and it also shows that a high IL-6 is related to low job control in men.
